# Species Composition, Diversity, and Biomass Estimation in Coastal and Marine Protected Areas of Terengganu, Peninsular Malaysia

**Elizabeth Pesiu** [1,*], **Gaik Ee Lee** [1,2,*], **Muhammad Razali Salam** [1], **Jamilah Mohd Salim** [1], **Kah Hoo Lau** [3], **Jean Wan Hong Yong** [4] and **Mohd Tajuddin Abdullah** [5,6]

1 Faculty of Science and Marine Environment, Universiti Malaysia Terengganu, Kuala Nerus 21030, Terengganu, Malaysia
2 Institute of Tropical Biodiversity and Sustainable Development, Universiti Malaysia Terengganu, Kuala Nerus 21030, Terengganu, Malaysia
3 Forest Biodiversity Division, Forest Research Institute Malaysia, Kepong 52109, Selangor, Malaysia
4 Department of Biosystems and Technology, Swedish University of Agricultural Sciences, 23053 Alnarp, Sweden
5 Faculty of Fisheries and Food Science, Universiti Malaysia Terengganu, Kuala Nerus 21030, Terengganu, Malaysia
6 West Wing, Menara Matrade, Jalan Sultan Haji Ahmad Shah, Kuala Lumpur 50480, Selangor, Malaysia
* Correspondence: elizabethpesiu@gmail.com (E.P.); gaik.ee@umt.edu.my (G.E.L.)

**Abstract:** We investigated and compared the tree species composition and diversity of different forest types in Setiu Wetlands and on the three major islands of Terengganu. A total of 24 plots of 25 m × 25 m with four plots in each study site were established, viz. *Melaleuca* swamp forest in Kampung Fikri, freshwater swamp forest in Kampung Gong Batu, mangrove forest in UMT Setiu research station, and the islands, namely Pulau Bidong, Pulau Redang, and Pulau Perhentian. We calculated the basal area, stand density, Importance Value Index, species diversity, and above-ground biomass in the designated study areas. We assessed 139 tree species from 96 genera and 50 families based on a total of 2608 tree samples of 5 cm DBH and above. The freshwater swamp forest harbored the highest number of species with 20 species in Setiu Wetlands, and among the islands, Pulau Redang had the highest with 56 species. *Melaleuca cajuputi* was the most dominant species in the *Melaleuca* swamp forest, while *Alstonia spatulata* and *Rhizophora apiculata* are expected in the freshwater swamp and mangrove forest, respectively. Pulau Bidong, Pulau Redang, and Pulau Perhentian are mostly represented by *Licania splendens*, *Shorea glauca*, and *Vatica* sp., respectively. All the dominant species but *Licania splendens* contributed to the highest amount of above-ground biomass. Our current study indicated that different forest types vary in composition and structure, which may contribute to their unique ecological roles within their specific environment.

**Keywords:** coastal and insular vegetation; protected areas; coastal wetland; freshwater swamp; mangrove; lowland dipterocarp forest



## 1. Introduction

The forest cover of Terengganu on the east coast of Peninsular Malaysia is still relatively extensive, with ca. 654,000 ha of forested areas [1]. Within these areas, 30,000 ha have been gazetted as a state park in Kenyir and ca. 1600 ha in Setiu Wetlands, including the freshwater wetland of Tasik Berombak [2,3]. National or state parks are thus far considered the best protected areas that do not allow logging or monoculture plantations. On the other hand, ca. 544,000 ha have been gazetted as Permanent Reserved Forests, which are still partly subjected to state forest management for timber production [1]. There are 45 forest reserves in Terengganu, consisting mainly of dipterocarp forests. Among these are Besul, Bukit Bauk, Bukit Kesing, Bukit Terendak, Cerul, Gunung Tebu, Hulu Besut, Hulu

Telemong, Hulu Terengganu, Jengai, Jerangau, Merchang, Pasir Raja Barat and Selatan, Pelagat, Petuang, Rasau Kerteh, Sungai Nipah, and Tembat, each with above 5000 ha [1]. Furthermore, there are 13 marine parks in Terengganu with 36 sea turtle nesting grounds, the largest amount in Peninsular Malaysia [4]. Thus, Terengganu's islands and the long coastline in Setiu Wetlands have been proposed as important marine ecological corridors: the Northern Terengganu Marine Park Islands-Setiu Wetlands Ecological Corridor of Peninsular Malaysia (Figure 1).

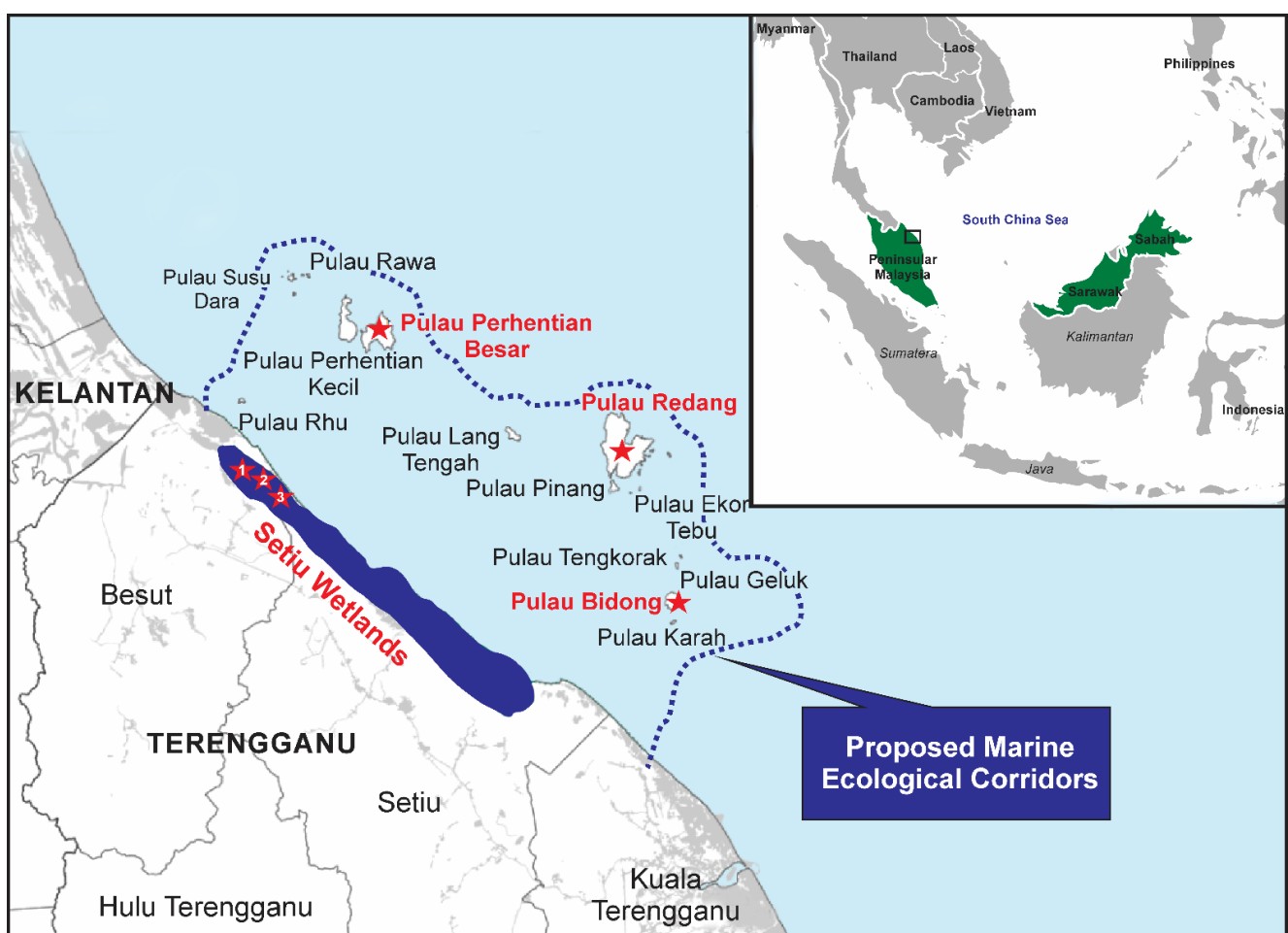

**Figure 1.** Map showing the Setiu Wetlands and islands of Terengganu. Red stars and bold red texts indicate the location of the study areas. Red star 1: mangrove in UMT research station, 2: freshwater swamp in Gong Batu, 3: *Melaleuca* swamp in Kampung Fikri. The dotted blue line is the proposed Northern Terengganu Marine Park Islands-Setiu Wetlands Ecological Corridor. Map modified from the 6th National Report of Malaysia to the Convention on Biological Diversity.

The Setiu Wetlands are a complex and heterogeneous landscape containing several different habitat types ranging from coastal forests to mangrove estuaries and freshwater swamps to hill dipterocarp forests [5,6]. Wetlands provide basic ecosystem regulatory services such as coastal protection [7,8], flood mitigation and erosion control [9,10], and nurseries for juvenile marine fishes and also support traditional livelihoods for the local population [6]. The wetlands are located northeast of Peninsular Malaysia in Terengganu and form part of the Setiu river basin. Much of the natural vegetation in Setiu Wetlands is still well-represented. These wetlands are considered the largest and possibly the most intact coastal wetland complex on the east coast of Peninsular Malaysia [2,6]. Due to their geographic isolation from the source population on the mainland, islands have long been known to support a set of unique populations, communities, and ecosystems. These

resistance entities have adapted through a geographical limitation with vital processes, properties, and interactions that occur in a simpler way [11]. The forests in coastal areas bring about ecological and socio-economic importance in terms of goods and services (such as forestry and fisheries resources; recreation and ecotourism) [12].

Assessment and monitoring of forest biological diversity in coastal areas can provide insight into conservation value and are essential for sustainable forest management. According to Noss [13], the challenge lies in defining sound and practical biodiversity monitoring systems that deliver scientific data to inform sustainable forest management. The use of inventories to determine biodiversity (composition, diversity of tree species, and frequency) is a common way to gather information for forest management operations [14]. On the other hand, the different types of forest types occurring in Terengganu may act as carbon sources and carbon sinks. Biologically, estimates of above-ground biomass are essential for studies of carbon stocks and the effects of deforestation and carbon sequestration on the global carbon balance [15,16]. However, such data are lacking for coastal areas in Terengganu.

Currently, published data analyses of various forest types in Setiu Wetlands and major islands in Terengganu are still lacking. Although much research has been conducted independently to record plant species, e.g., [17–19], the forest structure and floristic composition are still unknown. Most of the available literature only focused on qualitative measures attempting to document the absence or presence of tree species but not the physical structures, such as quantitative measures of species diversity within a plot, the numbers of individuals within a species, and comparison among forest types in each species. Therefore, we aimed to investigate and compare the tree species composition, diversity, and above-ground biomass of three different forest types in Setiu Wetlands: *Melaleuca* swamp, freshwater swamp, and mangrove forests. Furthermore, three marine parks, namely Pulau Bidong, Pulau Perhentian, and Pulau Redang, were also included in the present study. We hypothesize that the tree species diversity in different habitats in Setiu Wetlands varies according to soil and environment factors and changes along the coastline towards the inland of Setiu.

## 2. Materials and Methods

The study sites are in Setiu Wetlands State Park and on three major islands of Terengganu, i.e., Pulau Bidong, Pulau Redang, and Pulau Perhentian (Figure 1). Three different forest types, i.e., *Melaleuca* swamp, freshwater swamp, and mangrove forests in Setiu Wetlands, were selected in the present study (Figures 2 and 3). The islands of Terengganu are generally made up of lowland dipterocarp forests.

We used plot-based tree inventories to test the hypothesis. A total of 24 plots of 25 m × 25 m were established in all the study sites, with four plots in each of the six study sites. All trees with 5 cm diameter breast height (DBH) and above were surveyed, measured, and tagged with flagging ribbons. Trees were identified in-situ, while voucher specimens were collected for further identification and verification purposes. Specimens were deposited at the herbarium of Universiti Malaysia Terengganu (UMTP). Tree species were identified by E. Pesiu and M.R. Salam using the collection at UMTP and herbarium of Forest Research Institute Malaysia (KEP).

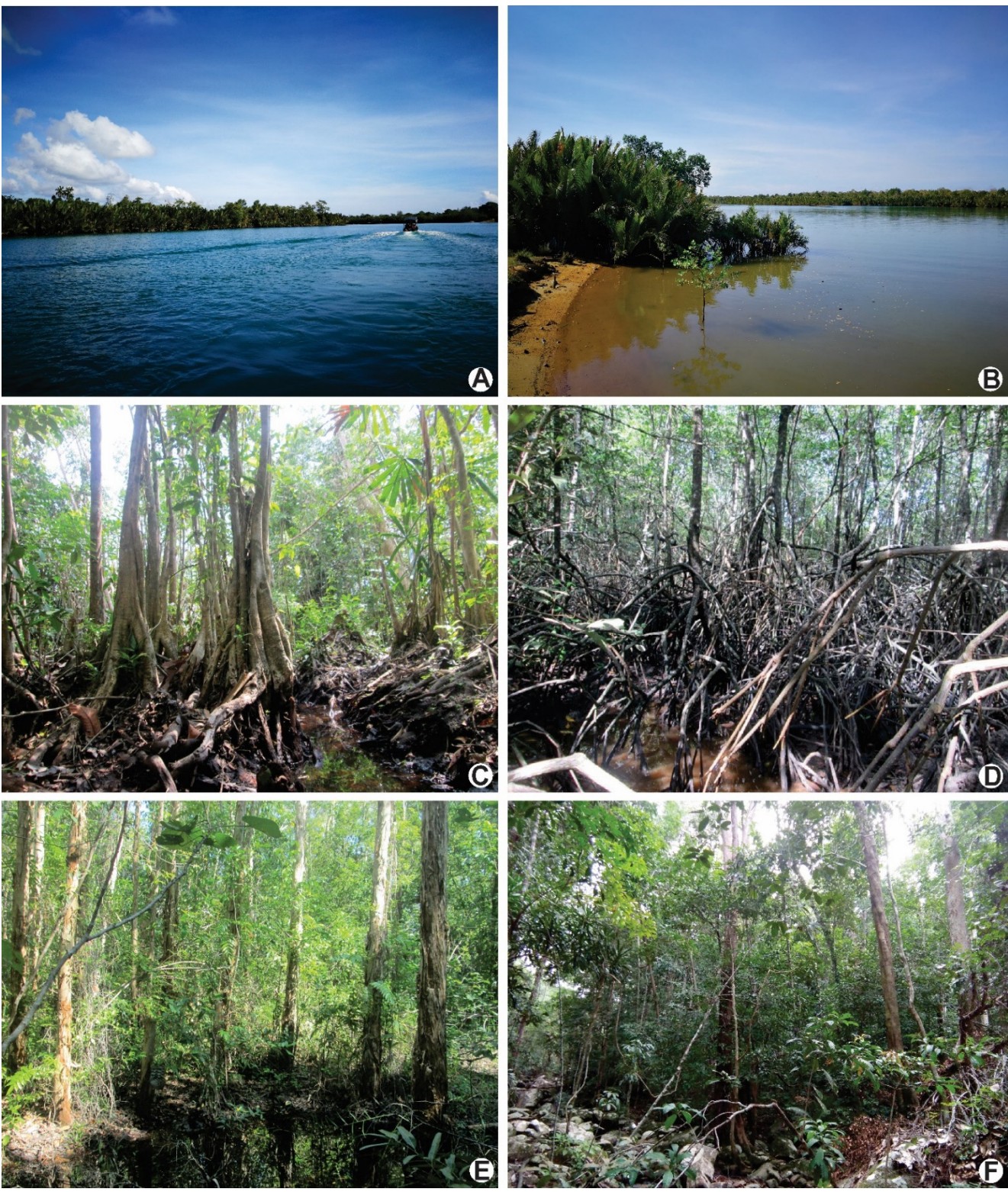

**Figure 2.** Different habitats at the study areas. (**A**,**B**) Setiu River, one of the four main rivers in Setiu Wetlands; (**C**) freshwater swamp forest; (**D**) mangrove forest; (**E**) *Melaleuca* swamp forest, (**F**) lowland dipterocarp forest in Pulau Redang.

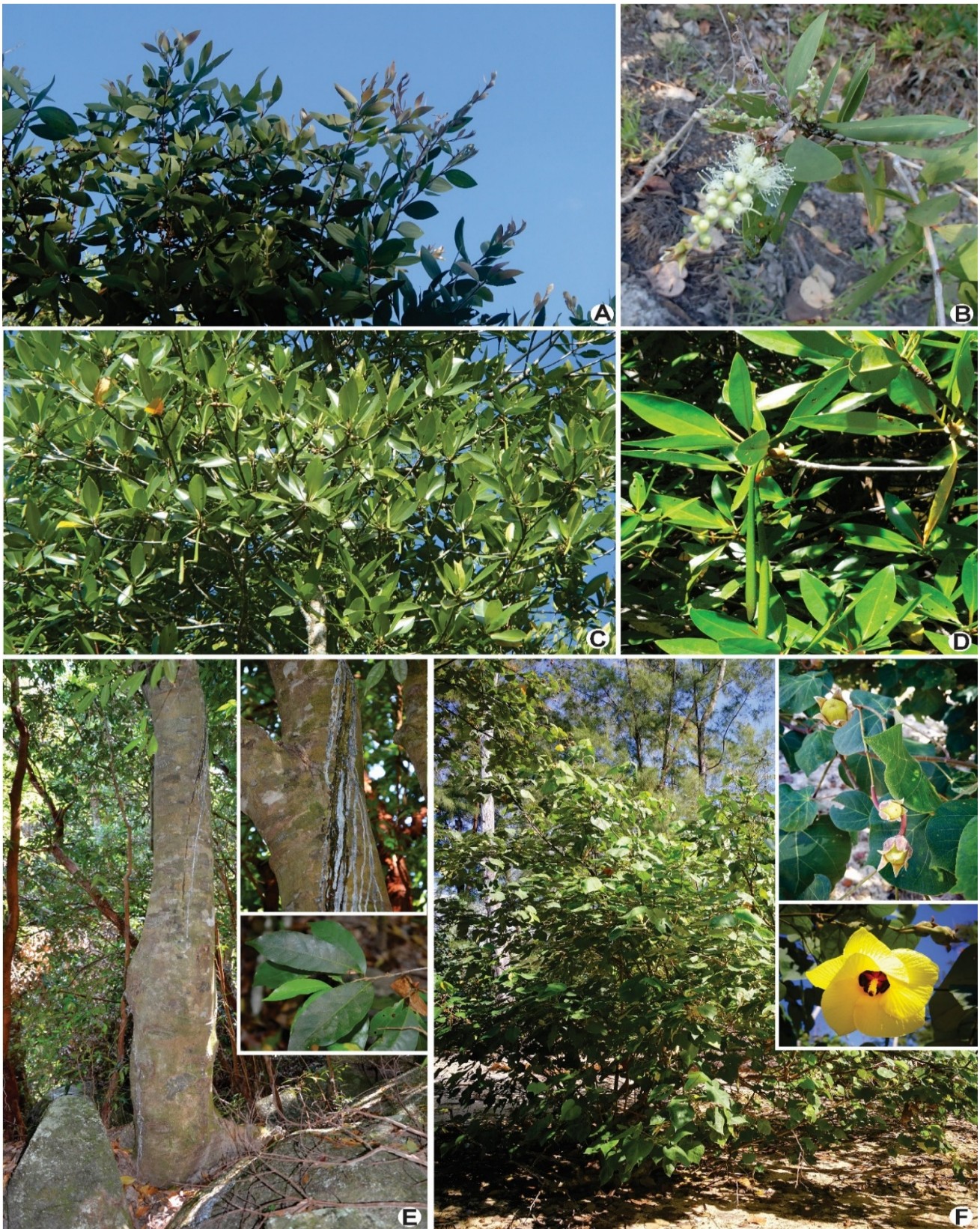

**Figure 3.** Some of the plants found in the study areas. (**A**) Top canopy of *Melaleuca cajuputi*; (**B**) leaves and inflorescence of *Melaleuca cajuputi*; (**C**) top canopy of *Rhizophora apiculata*; (**D**) leaves and propagules of *Rhizophora apiculata*; (**E**) bark of *Vatica* sp. (inset top: white resin, bottom: leaf arrangement); (**F**) habitat of *Talipariti tiliaceus* (inset top: fruit, bottom: flower).

We calculated the basal area, stand density, Importance Value Index, species diversity, and above-ground biomass in the designated study areas. The sample-based rarefaction curve [20] was used to compare the species abundance in different forest types at each study site. The 95% confidence intervals were constructed where samples were drawn without replacement and calculated as ±2 standard deviations from the expected values [21]. Data was computed by using PAST Version 3.0 software [22]. Community composition within the plots was carried out with several abundance parameters such as basal area ($m^2$/ha), stand density (Ind/ha) and Importance Value index ($IV_i$). The $IV_i$ was calculated by summing the values of relative density (RD), relative dominance (based on basal area) (RB), and relative frequency (RF) of each species or family ($IV_i$ = RD + RB + RF) [23]. Species diversity was determined using the Shannon Diversity Index (H′) [24] as follows:

$$H = \sum_{i=1}^{s} -(Pi * \ln Pi)$$

where s is the number of species; Pi is the proportion of individuals or the abundance of the ith species expressed as a proportion of total abundance. The estimation of above-ground biomass was calculated following Kato et al. [25]. The equation takes into account of the stem biomass ($W_s$), branches biomass ($W_b$) and leaves biomass ($W_i$). The above-ground biomass is estimated by adding up the value of stem, branches, and leaves biomass. The equations are as follows:

$$\text{Stem biomass } (W_s) = 0.313 \, (DBH^2 \, H)^{0.9733}$$

$$\text{Branches biomass } (W_b) = 0.136 \, W_s^{1.041}$$

$$\text{Leaves biomass } (W_i) = \frac{125 \times 0.124 \, W_s^{0.794}}{0.124 \, W_s^{0.794} + 125}$$

where,

$$\text{Tree height, } H = \frac{(122 \times DBH)}{(2 \, DBH + 61)}$$

$$\text{Above-ground biomass (kg)} = W_s + W_b + W_i$$

## 3. Results

### 3.1. Species Richness and Diversity

We assessed 139 tree species from 96 genera and 50 families. This sampling was based on a total of 2608 tree samples of 5 cm DBH and above (Tables 1 and 2). The detail of the tree species is shown in Table 2. In Setiu Wetlands, the freshwater swamp forest showed the highest number of species with 20 species compared to the *Melaleuca* swamp and mangrove forest with 13 species each. On the islands, Pulau Redang recorded the highest number of species with 56 species, followed by Pulau Bidong and Pulau Perhentian, with 55 and 20 species, respectively. In terms of density, freshwater swamp forest had the highest density and largest basal area with 1992 ind/ha and 17.38 $m^2$/ha; among the islands, Pulau Bidong had the highest density with 2700 ind/ha and Pulau Redang had the largest basal area with 52.09 $m^2$/ha (Table 1). The rarefaction curves differed significantly, and 95% confidence intervals did not overlap among the six study areas (Figures S1 and S2). All the curves are still in the exponential phase and are non-asymptotic with currently available sample sizes.

**Table 1.** Comparison of basic tree inventory metrics and the total above-ground biomass in each study site.

| | Setiu Wetlands | | | Islands | | |
|---|---|---|---|---|---|---|
| | *Melaleuca* Swamp | Freshwater Swamp | Mangrove | Pulau Bidong | Pulau Redang | Pulau Perhentian |
| Number of families | 13 | 17 | 10 | 28 | 27 | 13 |
| Number of genera | 14 | 20 | 13 | 43 | 42 | 14 |
| Number of species | 13 | 20 | 13 | 55 | 56 | 20 |
| Stem density (trees DBH $\geq$ 5 cm DBH, trees/ha) | 1432 | 1992 | 1208 | 2700 | 1088 | 1500 |
| Basal area of trees (m$^2$/ha) | 15.22 | 17.38 | 8.84 | 25.04 | 52.09 | 24.64 |
| Total above-ground biomass (t/ha) | 130.92 | 147.98 | 69.47 | 216.73 | 728.04 | 244.05 |

**Table 2.** Species list based on plot-based tree inventories. MS = *Melaleuca* swamp, FS = freshwater swamp, M = mangrove in Setiu Wetlands; PB = Pulau Bidong, PR = Pulau Redang, PP = Pulau Perhentian.

| Family | Species | MS | FS | M | PB | PR | PP |
|---|---|---|---|---|---|---|---|
| Anacardiaceae | *Bouea oppositifolia* | − | − | − | + | + | − |
| | *Bouea* sp. | − | − | − | − | + | − |
| | *Buchanania arborescens* | − | − | − | + | − | − |
| | *Campnosperma coriaceum* | − | + | − | − | − | − |
| | *Dracontomelon dao* | − | − | − | + | − | − |
| | *Mangifera macrocarpa* | − | − | − | − | + | − |
| | *Mangifera odorata* | − | − | − | − | + | − |
| | *Parishia insignis* | − | − | − | − | + | − |
| | *Swintonia floribunda* | − | − | − | − | + | − |
| | *Swintonia schwenkii* | − | − | − | + | − | − |
| Annonaceae | *Goniothalamus tenuifolius* | − | − | − | − | + | − |
| | *Polyalthia sumatrana* | − | − | − | − | + | − |
| Apocynaceae | *Alstonia spatulata* | − | + | − | − | − | − |
| Aquifoliaceae | *Ilex cymosa* | + | + | + | − | − | − |
| Asparagaceae | *Dracaena maingayi* | − | − | − | − | + | − |
| Bignoniaceae | *Dolichandrone spathacea* | − | − | + | − | − | − |
| Burseraceae | *Dacryodes rostrata* | − | − | − | − | + | − |
| | *Santiria rubiginosa* | − | − | − | − | + | − |
| | *Santiria* sp. | − | − | − | + | − | − |
| Calophyllaceae | *Calophyllum rupicola* | − | − | − | + | − | + |
| | *Mesua ferrea* | − | − | − | − | + | − |
| | *Mesua lepidota* | − | − | − | + | + | − |
| Cannabaceae | *Gironniera* sp. | − | − | − | + | − | − |
| Celastraceae | *Euonymus cochinchinensis* | − | − | − | − | − | − |
| | *Kokoona sessilis* | − | − | − | − | + | − |
| Chrysobalanaceae | *Licania splendens* | − | − | − | + | − | − |
| | *Maranthes corymbosa* | − | − | − | − | + | − |
| | *Maranthes* sp. | − | − | − | + | − | − |
| Clusiaceae | *Garcinia eugenifolia* | − | + | − | + | − | − |
| | *Garcinia hombroniana* | + | + | − | + | − | − |
| | *Garcinia nigrolineata* | − | − | − | + | − | − |
| | *Garcinia* sp. | − | − | − | − | + | − |
| Dipterocarpaceae | *Dipterocarpus chartaceus* | − | − | − | + | − | − |
| | *Shorea glauca* | − | − | − | − | + | − |
| | *Shorea materialis* | − | − | − | + | − | − |
| | *Vatica* sp. | − | − | − | + | + | + |
| Ebenaceae | *Diospyros pilosanthera* | − | − | − | − | + | − |
| | *Diospyros* sp. 1 | − | − | − | + | − | − |

**Table 2.** *Cont.*

| Family | Species | MS | FS | M | PB | PR | PP |
|---|---|---|---|---|---|---|---|
| | *Diospyros* sp. 2 | − | − | − | − | + | − |
| | *Diospyros* sp. 3 | − | − | − | − | + | − |
| Elaeocarpaceae | *Elaeocarpus macrocerus* | + | + | − | − | − | − |
| | *Elaeocarpus mastersii* | − | − | − | − | − | − |
| | *Elaeocarpus* sp. | − | − | − | − | − | + |
| Erythroxylaceae | *Erythroxylum cuneatum* | − | − | − | + | − | − |
| Euphorbiaceae | *Croton oblongus* | − | − | − | − | + | − |
| | *Drypetes* sp. | − | − | − | + | − | − |
| | *Excoecaria agallocha* | − | − | + | − | − | − |
| | *Koilodepas* sp. | − | − | − | − | + | − |
| | *Macaranga hypoleuca* | − | + | − | − | − | − |
| | *Suregada multiflora* | + | + | − | − | + | − |
| Fabaceae | *Acacia mangium* | − | − | + | − | − | − |
| | *Archidendron contortum* | − | − | − | + | − | − |
| | *Callerya atropurpurea* | − | − | − | − | − | − |
| | *Intsia bijuga* | − | − | + | − | − | − |
| | *Ormosia sumatrana* | − | − | − | − | + | − |
| | *Sindora cochinchinensis* | − | − | − | − | − | − |
| | *Lithocarpus rassa* | − | − | − | + | − | + |
| Flacourtiaceae | *Hydnocarpus* sp. | − | − | − | − | + | − |
| Hypericaceae | *Cratoxylum arborescens* | − | + | − | − | − | − |
| | *Cratoxylum formosum* | − | − | − | − | − | + |
| Lamiaceae | *Teijsmanniodendron coriaceum* | − | − | − | + | + | − |
| Lauraceae | *Cinnamomum sintoc* | − | − | − | − | + | − |
| | *Litsea* sp. | − | − | − | − | + | − |
| | *Neolitsea zeylanica* | + | + | − | − | − | − |
| Lecythidaceae | *Barringtonia macrostachya* | − | − | − | − | − | − |
| | *Barringtonia scortechinii* | − | − | − | − | + | − |
| Loganiaceae | *Norrisia malaccensis* | − | − | − | + | − | − |
| Malvaceae | *Heritiera littoralis* | − | − | + | − | − | − |
| | *Heritiera simplicifolia* | − | − | − | − | + | − |
| | *Heritiera* sp. 1 | − | − | − | + | − | − |
| | *Heritiera* sp. 2 | − | − | − | + | − | − |
| | *Talipariti tiliaceus* | + | + | + | − | − | − |
| Melastomataceae | *Memecylon edule* | + | − | − | − | − | − |
| | *Memecylon* sp. | − | − | − | + | + | − |
| Meliaceae | *Sandoricum koetjape* | − | − | − | − | − | − |
| | *Xylocarpus rumphii* | − | − | + | − | − | − |
| Moraceae | *Artocarpus kemando* | − | − | − | − | + | − |
| | *Artocarpus lanceifolius* | − | − | − | − | + | − |
| | *Artocarpus scortechinii* | − | − | − | − | − | − |
| | *Ficus* sp. | − | + | − | − | − | − |
| | *Streblus taxoides* | + | − | − | − | − | − |
| Myricaceae | *Myrica esculenta* | + | + | − | − | − | − |
| Myristicaceae | *Knema glauca* | − | − | − | + | + | − |
| | *Knema globularia* | − | − | − | − | + | − |
| | *Knema laurina* | − | − | − | + | − | − |
| Myrtaceae | *Melaleuca cajuputi* | + | + | − | − | − | − |
| | *Rhodamnia cinerea* | − | − | − | + | − | + |
| | *Syzygium cerinum* | − | − | − | + | − | + |
| | *Syzygium cinereum* | − | − | − | + | − | − |
| | *Syzygium grande* | − | − | − | + | + | − |
| | *Syzygium* sp. 1 | − | − | − | + | + | + |
| | *Syzygium* sp. 2 | − | − | − | + | − | + |
| | *Syzygium* sp. 3 | − | − | − | + | − | + |
| | *Syzygium* sp. 4 | − | − | − | + | − | + |
| | *Syzygium* sp. 5 | − | − | − | − | − | + |

**Table 2.** *Cont.*

| Family | Species | MS | FS | M | PB | PR | PP |
|---|---|---|---|---|---|---|---|
| | *Syzygium* sp. 6 | – | – | – | – | – | + |
| | *Syzygium syzygioides* | – | – | – | + | – | – |
| | *Syzygium zeylanicum* | – | + | – | + | + | – |
| Ochnaceae | *Brackenridgea hookeri* | – | + | – | + | – | – |
| | *Campylospermum serratum* | – | – | – | + | – | – |
| Pentaphylacaceae | *Adinandra dumosa* | – | – | – | – | – | – |
| Peraceae | *Chaetocarpus castanocarpus* | – | – | – | + | – | + |
| Phyllanthaceae | *Baccaurea parviflora* | – | – | – | – | + | – |
| | *Cleistanthus* sp. 1 | – | – | – | – | + | – |
| | *Cleistanthus* sp. 2 | – | – | – | – | + | – |
| | *Cleistanthus sumatranus* | – | – | – | – | + | – |
| | *Glochidion* sp. | – | – | + | – | – | – |
| Picrodendraceae | *Austrobuxus nitidus* | – | – | – | + | – | – |
| Pittosporaceae | *Pittosporum ferrugineum* | + | + | – | – | – | – |
| Polygalaceae | *Xanthophyllum* sp. | – | – | – | + | – | – |
| Primulaceae | *Rapanea porteriana* | – | – | – | – | – | + |
| Rhizophoraceae | *Bruguiera gymnorrhiza* | – | – | + | – | – | – |
| | *Gynotroches axillaris* | – | + | – | – | – | – |
| | *Rhizophora apiculata* | – | – | + | – | – | – |
| Rubiaceae | *Aidia wallichiana* | – | – | – | – | + | – |
| | *Canthium glabrum* | – | – | – | – | + | – |
| | *Canthium nitidum* | – | – | – | + | – | – |
| | *Canthium* sp. | – | – | – | – | + | – |
| | *Diplospora malaccensis* | – | – | – | + | + | – |
| | *Ixora pendula* | – | – | – | – | + | – |
| | *Morinda elliptica* | – | – | – | + | – | – |
| | *Psydrax* sp. | – | – | – | + | – | + |
| | *Timonius wallichianus* | – | – | – | + | – | – |
| Rutaceae | *Acronychia pedunculata* | – | – | – | + | + | – |
| | *Atalantia monophylla* | – | – | – | – | + | – |
| Sapindaceae | *Guioa bijuga* | – | – | + | + | – | + |
| | *Nephelium* sp. | – | – | – | – | – | – |
| Sapotaceae | *Madhuca longifolia* | – | – | – | + | – | – |
| | *Madhuca sericea* | – | – | – | – | + | – |
| | *Madhuca tubulosa* | – | – | – | – | + | – |
| | *Palaquium obovatum* | – | – | – | + | – | + |
| | *Pouteria malaccensis* | – | – | – | – | + | – |
| | *Pouteria obovata* | + | + | + | + | + | – |
| Simaroubaceae | *Eurycoma longifolia* | – | – | – | + | – | – |
| Sterculiaceae | *Sterculia parviflora* | – | – | – | – | + | – |
| Symplocaceae | *Symplocos adenophylla* | – | – | – | + | – | + |
| Verbenaceae | *Vitex pinnata* | – | – | – | – | – | + |
| | *Vitex trifolia* | + | + | – | – | – | – |
| Violaceae | *Rinorea* sp. | – | – | – | – | + | – |
| Total: 50 families | 139 species | 13 | 20 | 13 | 55 | 56 | 20 |

In Setiu Wetlands, the Shannon–Wiener Diversity Index (H′) shows that freshwater swamp has the highest tree diversity (Table 3). On the other hand, the diversity index on islands were slightly higher, of which Pulau Bidong recorded the highest with 3.22 (Table 3). As for species evenness, the forests indicated low evenness. Pulau Perhentian had the highest evenness with 0.68, while Pulau Bidong and Pulau Redang indicated moderate evenness.

Regarding DBH size classification, the study areas in Setiu Wetlands showed a low number of diameter class distributions with only 2–3 main diameter classes (Figure 4). Most of the trees within the plot were small to medium in size, and the majority were in the 5–14.99 cm DBH class, followed by 15–24.9 cm, and only a few in the 25–34.9 cm. In the

*Melaleuca* swamp forest, *Melaleuca cajuputi* from the family Myrtaceae dominated the forest plot, while in the mangrove and freshwater swamp, *Rhizophora apiculata* (Rhizophoraceae) and *Alstonia spatulata* (Apocynaceae) recorded the highest, respectively.

**Table 3.** The Shannon–Wiener Diversity Index (H′) and species evenness (E) in Setiu Wetlands and three major islands of Terengganu.

| Study Site | Shannon Index (H′) | Evenness (E) |
|---|---|---|
| Setiu Wetlands: | | |
| Freshwater swamp forest | 1.85 | 0.38 |
| *Melaleuca* swamp forest | 1.67 | 0.30 |
| Mangrove forest | 0.98 | 0.20 |
| Islands: | | |
| Pulau Bidong | 3.22 | 0.46 |
| Pulau Redang | 3.18 | 0.43 |
| Pulau Perhentian | 1.95 | 0.68 |

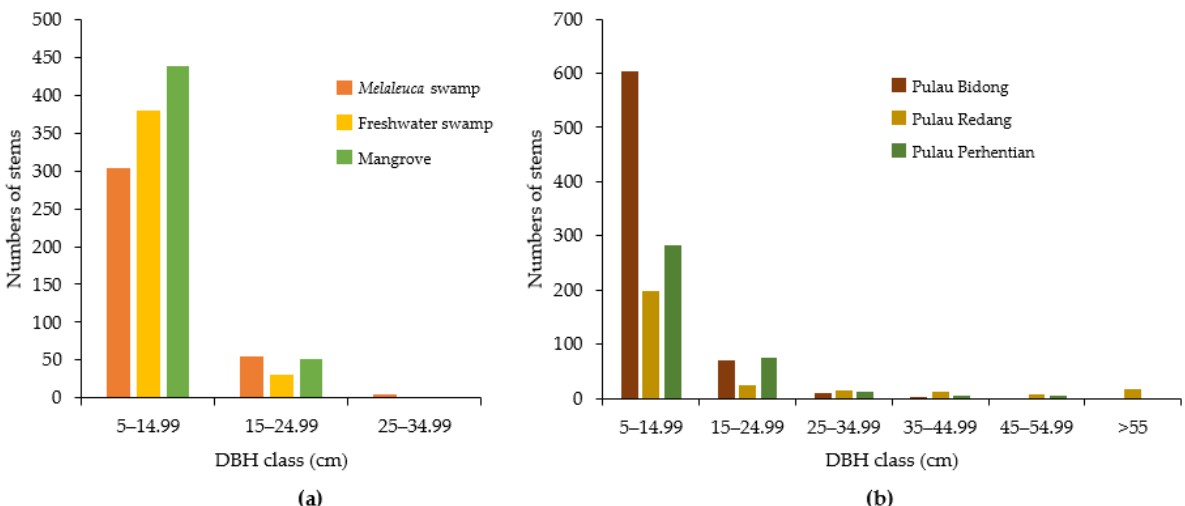

**Figure 4.** Diameter class distribution. (**a**) The *Melaleuca* swamp, freshwater swamp and mangrove forests in Setiu Wetlands; (**b**) the three major islands of Terengganu, Pulau Bidong, Pulau Redang, and Pulau Perhentian.

The other families were scattered, such as Aquifoliaceae, Malvaceae, Sapotaceae, Pittosporaceae, Euphorbiaceae, Verbenaceae, Anacardiaceae, and Fabaceae. Myrtaceae had the largest basal area, particularly in the *Melaleuca* swamp, followed by the Aquifoliaceae, Sapotaceae, and Malvaceae. The abundance of the family Rhizophoraceae with the most dominant species, *Rhizophora apiculata*, formed the major structure of the mangrove forest with a total basal area of 7.08 m$^2$ ha$^{-1}$. In the freshwater swamp, the family Apocynaceae (*Alstonia spatulata*) recorded the largest basal area with a total of 9.77 m$^2$ ha$^{-1}$.

From the islands, the study areas demonstrated more DBH classes of at least 4–6 classes of diameter distributions (Figure 4). Most of the trees within the plot were small to medium in size, and the majority were in the 5–14.99 cm DBH class, followed by 15–24.9 cm and 25–34.9 cm. In Pulau Bidong and Pulau Perhentian, the family Myrtaceae and genus *Syzygium* showed the highest number of individuals in diameter class distribution. However, the species varies accordingly based on the diameter size. In Pulau Bidong, *Licania splendens* tends to be abundant at small sizes within the forest, while *Lithocarpus rassa* and *Syzygium cinereum* were found growing in medium to bigger sizes. In Pulau Perhentian, *Vatica* sp. was dominant in small and larger diameter sizes, while *Palaquium obovatum* and *Buchanania arborescens* were in more medium to large sizes. In Pulau Redang, *Diospyros pilosanthera* from the family Ebenaceae recorded the highest at 5–14.99 diameter. Meanwhile, *Madhuca*

*sericea* from the family Sapotaceae was dominant at 15–24.99 diameter class. For the diameter class of 25 cm DBH and above, *Shorea glauca* from the family Dipterocarpaceae recorded the highest.

In terms of density, the family Myrtaceae showed the highest in Pulau Bidong and Pulau Perhentian, with 572 individual/ha and 444 individual/ha, respectively. The family Dipterocarpaceae was the second highest and thrived well on all three islands. Each site had 384 ind/ha, 172 ind/ha, and 280 ind/ha in Pulau Bidong, Pulau Redang, and Pulau Perhentian, respectively. Furthermore, in the basal area, the family Dipterocarpaceae showed the highest in Pulau Redang with 35.49 $m^2$/ha, and Myrtaceae in Pulau Perhentian with 9.58 $m^2$/ha and Pulau Bidong with 7.09 $m^2$/ha. *Shorea glauca* from the family Dipterocarpaceae in Pulau Redang showed the largest basal area with 35.28 $m^2$/ha while *Vatica* sp. in Pulau Perhentian and *Syzygium cinereum* in Pulau Bidong, each with 5.04 $m^2$/ha and 2.68 $m^2$/ha respectively.

### 3.2. Importance Value Index

In the *Melaleuca* swamp forest, *Melaleuca cajuputi* had the highest importance value ($IV_i$ = 112.52) (Table 4). The other dominant species within the same forest type includes *Ilex cymosa* ($IV_i$ = 62.72), *Talipariti tiliaceus* ($IV_i$ = 23.18), and *Pouteria obovata* ($IV_i$ = 19.09). For the mangrove forest, the most dominant species were the *Rhizophora apiculata* ($IV_i$ = 173.67), followed by *Excoecaria agallocha* ($IV_i$ = 22.60), *Pouteria obovata* ($IV_i$ = 20.84), *Talipariti tiliaceus* ($IV_i$ = 15.20), *Intsia bijuga* ($IV_i$ = 14.97), and *Heritiera littoralis* ($IV_i$ = 18.82). The least dominant species was *Dolichandrone spathaceae* with $IV_i$ of 3.75. The most dominant species in freshwater swamp forest was *Alstonia spatulata* with $IV_i$ of 105.96. The other dominant species were *Gynotroches axillaris* ($IV_i$ = 49.99), *Macaranga hypoleuca* ($IV_i$ = 33.91), *Vitex trifolia* ($IV_i$ = 17.40), and *Campnosperma coriaceum* ($IV_i$ = 13.49).

**Table 4.** Relative dominance ($RD_i$), relative coverage ($RC_i$), relative frequency ($Rf_i$), and importance value ($IV_i$) of tree species with DBH of 5 cm in Setiu Wetlands and three major islands of Terengganu.

| Species | $RD_i$ | $RC_i$ | $Rf_i$ | $IV_i$ |
|---|---|---|---|---|
| **Setiu Wetlands: *Melaleuca* swamp forest** | | | | |
| *Melaleuca cajuputi* | 56.72 | 44.69 | 11.11 | 112.52 |
| *Ilex cymosa* | 25.07 | 26.54 | 11.11 | 62.72 |
| *Talipariti tiliaceus* | 4.53 | 7.54 | 11.11 | 23.18 |
| *Pouteria obovata* | 3.23 | 4.75 | 11.11 | 19.09 |
| *Neolitsea zeylanica* | 3.15 | 3.63 | 11.11 | 17.89 |
| *Suregada multiflora* | 1.98 | 2.23 | 11.11 | 15.33 |
| *Pittosporum ferrigeneum* | 2.48 | 4.19 | 5.56 | 12.23 |
| *Streblus taxoides* | 0.79 | 0.84 | 5.56 | 7.18 |
| *Myrica esculenta* | 0.32 | 0.56 | 5.56 | 6.43 |
| *Vitex trifolia* | 0.61 | 1.96 | 2.78 | 5.34 |
| *Suregada multiflora* | 1.98 | 2.23 | 11.11 | 15.33 |
| **Mangrove forest** | | | | |
| *Rhizophora apiculata* | 81.83 | 78.05 | 13.79 | 173.67 |
| *Excoecaria agallocha* | 6.89 | 5.37 | 10.34 | 22.6 |
| *Pouteria obovata* | 2.41 | 4.63 | 13.79 | 20.84 |
| *Talipariti tiliaceus* | 1.68 | 3.17 | 10.34 | 15.2 |
| *Intsia bijuga* | 1.94 | 2.68 | 10.34 | 14.97 |
| *Heritiera littoralis* | 2.04 | 2.44 | 10.34 | 14.82 |
| *Guioa bijuga* | 0.70 | 1.22 | 6.90 | 8.82 |
| *Ilex cymosa* | 1.13 | 0.49 | 6.90 | 8.51 |
| *Bruguiera gymnorhiza* | 0.55 | 0.73 | 3.45 | 4.73 |
| *Acacia mangium* | 0.47 | 0.24 | 3.45 | 4.16 |

**Table 4.** *Cont.*

| Species | RD$_i$ | RC$_i$ | Rf$_i$ | IV$_i$ |
|---|---|---|---|---|
| **Freshwater swamp forest** | | | | |
| *Alstonia spatulata* | 56.1 | 41.16 | 8.70 | 105.96 |
| *Gynotroches axillaris* | 17.0 | 24.30 | 8.70 | 49.99 |
| *Macaranga hypoleuca* | 11.96 | 13.25 | 8.70 | 33.91 |
| *Vitex trifolia* | 3.28 | 5.42 | 8.70 | 17.40 |
| *Campnosperma coriaceum* | 1.99 | 2.81 | 8.70 | 13.49 |
| *Cratoxylum arborescens* | 1.07 | 1.41 | 6.52 | 8.99 |
| *Melaleuca cajuputi* | 2.32 | 1.20 | 4.35 | 7.88 |
| *Ilex cymosa* | 1.28 | 1.61 | 4.35 | 7.24 |
| *Pouteria obovata* | 0.72 | 0.80 | 4.35 | 5.87 |
| *Brackenridgea hookeri* | 0.61 | 0.80 | 4.35 | 5.76 |
| **Islands: Pulau Bidong** | | | | |
| *Licania splendens* | 10.58 | 3.17 | 12.89 | 26.64 |
| *Calophyllum rupicola* | 9.75 | 3.17 | 9.63 | 22.55 |
| *Vatica* sp. | 8.32 | 3.17 | 10.96 | 22.46 |
| *Syzygium cinereum* | 10.78 | 3.17 | 7.26 | 21.21 |
| *Lithocarpus rassa* | 10.09 | 3.17 | 3.56 | 16.82 |
| *Syzygium cerinum* | 4.71 | 3.17 | 4.44 | 12.33 |
| *Syzygium* sp. 2 | 5.94 | 2.38 | 3.85 | 12.17 |
| *Austrobuxus nitidus* | 3.80 | 3.17 | 4.74 | 11.72 |
| *Symplocos adenophylla* | 2.32 | 3.17 | 5.33 | 10.83 |
| *Buchanania arborescens* | 3.22 | 3.17 | 3.11 | 9.51 |
| **Pulau Redang** | | | | |
| *Shorea glauca* | 67.77 | 15.19 | 4.49 | 87.45 |
| *Madhuca sericea* | 4.53 | 12.22 | 4.49 | 21.25 |
| *Diospyros pilosanthera* | 0.67 | 9.26 | 4.49 | 14.42 |
| *Diospyros* sp. | 0.91 | 10 | 3.37 | 14.28 |
| *Litsea* sp. | 0.90 | 7.41 | 4.49 | 12.81 |
| *Barringtonia scortechinii* | 0.98 | 4.44 | 3.37 | 8.80 |
| *Artocarpus lanceifolius* | 3.69 | 1.85 | 2.25 | 7.79 |
| *Cleistanthus sumatranus* | 0.96 | 3.33 | 3.37 | 7.66 |
| *Koilodepas* sp. | 0.21 | 3.33 | 3.37 | 6.91 |
| *Swintonia floribunda* | 4.01 | 0.37 | 1.12 | 5.50 |
| **Pulau Perhentian** | | | | |
| *Vatica* sp. | 20.46 | 18.93 | 6.78 | 46.18 |
| *Syzygium cerina* | 13.28 | 16.80 | 6.78 | 36.86 |
| *Buchanania arborescens* | 12.91 | 12.00 | 6.78 | 31.69 |
| *Symplocos adenophylla* | 7.41 | 9.87 | 6.78 | 24.06 |
| *Syzygium* sp. 1 | 5.46 | 6.67 | 6.78 | 18.91 |
| *Psydrax* sp. | 3.28 | 8.80 | 5.08 | 17.16 |
| *Chaetocarpus castanocarpus* | 3.66 | 5.87 | 6.78 | 16.31 |
| *Calophyllum rupicola* | 5.43 | 5.60 | 5.08 | 16.12 |
| *Syzygium* sp. 3 | 4.54 | 2.40 | 5.08 | 12.03 |
| *Syzygium* sp. 4 | 5.67 | 1.07 | 3.39 | 10.13 |

In Pulau Bidong, *Licania splendens* was the most dominant species within the plot, having the highest importance value (IV$_i$ = 26.64). The least dominant species were *Timonius wallichianus*, *Memecylon* sp., *Diplospora malaccensis*, and *Garcinia eugenifolia* with IV$_i$ of 0.98. The most dominant species in Pulau Redang was *Shorea glauca* (IV$_i$ = 87.85), followed by *Madhuca sericea* (IV$_i$ = 21.25), *Diospyros pilosanthera* (IV$_i$ = 14.42), *Diospyros* sp. 1 (IV$_i$ = 14.28), and *Litsea* sp. (IV$_i$ = 12.81). Meanwhile, the least dominant species were *Mangifera odorata*, *Cleistanthus* sp. 1, *Diospyros* sp. 2, *Suregada multiflora*, *Madhuca tubulosa*, *Dracaena maingayi*, *Memecylon* sp., *Ormosia sumatrana*, and *Syzygium zeylanicum* with IV$_i$ of 1.51. In Pulau

Perhentian, the most dominant species was *Vatica* sp. ($IV_i = 46.18$), and the co-dominant species were *Syzygium cerina* ($IV_i = 36.86$) and *Buchanania arborescens* ($IV_i = 31.69$) (Table 4).

*3.3. Above-Ground Biomass Estimation*

The family Myrtaceae in Pulau Bidong and Pulau Perhentian recorded the highest above-ground biomass with a total of 67.07 t/ha and 100.32 t/ha, respectively, contributing to 30.95% and 41.11%. Meanwhile, in Pulau Redang, the family Dipterocarpaceae was the highest, with a total of 534.35 t/ha (73.40%). In Pulau Bidong, *Syzygium cinereum* contributed the most with a total of 26.49 t/ha (11.99%), while the *Shorea glauca* was the highest in Pulau Redang, 532.37 t/ha (73.10%). In Pulau Perhentian, *Vatica* sp. contributed the most, with an estimated total above-ground biomass of 49.71 t/ha (20.36%).

The total amount of above-ground biomass concerning the family differs according to each study site that follows the types of forest formation and vegetation. The family Myrtaceae was the highest for the Melaleuca swamp forest with 84.88 t/ha (65.01%) (Table 5). Meanwhile, in the mangrove forest, the family Rhizophoraceae contributed 55.70 t/ha (80.18%). The family Apocynaceae were the highest in the freshwater swamp forest with 87.40 t/ha or 58.06% of the total above-ground biomass. Generally, the families Myrtaceae and Dipterocarpaceae usually contributed a high amount of above-ground biomass in islands. In Pulau Bidong and Pulau Perhentian, the Myrtaceae was the highest, with a total amount of 67.07 t/ha and 100.32 t/ha contributing to 30.95% and 41.11% of the total above-ground biomass. Meanwhile, the family Dipterocarpaceae was the highest in Pulau Redang, with a total of 534.35 t/ha (73.40%).

**Table 5.** List of five species that contribute to the highest amount of above-ground biomass (AGB).

| Study Area | Species | Family | AGB (t/ha) | % |
|---|---|---|---|---|
| Setiu Wetlands: *Melaleuca* swamp forest | *Melaleuca cajuputi* | Myrtaceae | 84.88 | 64.83 |
| | *Ilex cymosa* | Aquifoliaceae | 28.10 | 21.46 |
| | *Pouteria obovata* | Sapotaceae | 4.17 | 3.18 |
| | *Pittosporum ferrigenium* | Pittosporaceae | 3.59 | 2.74 |
| | *Talipariti tiliaceus* | Malvaceae | 3.51 | 2.68 |
| Mangrove forest | *Rhizophora apiculata* | Rhizophoraceae | 55.70 | 80.17 |
| | *Excoecaria agallocha* | Euphorbiaceae | 5.65 | 8.13 |
| | *Heritiera littoralis* | Malvaceae | 2.01 | 2.89 |
| | *Pouteria obovata* | Sapotaceae | 1.79 | 2.58 |
| | *Intsia bijuga* | Fabaceae | 1.47 | 2.12 |
| Freshwater swamp forest | *Alstonia spatulata* | Apocynaceae | 87.27 | 58.97 |
| | *Gynotroches axillaris* | Rhizophoraceae | 22.79 | 15.40 |
| | *Macaranga hypoleuca* | Euphorbiaceae | 17.7 | 11.96 |
| | *Vitex trifolia* | Verbenaceae | 4.37 | 2.95 |
| | *Melaleuca cajuputi* | Myrtaceae | 3.55 | 2.40 |
| Islands: Pulau Bidong | *Syzygium cinereum* | Myrtaceae | 26.49 | 11.99 |
| | *Lithocarpus rassa* | Fagaceae | 25.18 | 11.62 |
| | *Licania splendens* | Chrysobalanaceae | 21.22 | 9.79 |
| | *Calophyllum rupicola* | Calophyllaceae | 20.09 | 9.27 |
| | *Vatica* sp. | Dipterocarpaceae | 16.21 | 7.48 |
| Pulau Redang | *Shorea glauca* | Dipterocarpaceae | 532.27 | 73.10 |
| | *Swintonia floribunda* | Anacardiaceae | 34.20 | 4.70 |
| | *Mangifera macrocarpa* | Anacardiaceae | 29.67 | 4.08 |
| | *Artocarpus lanceifolius* | Moraceae | 24.11 | 3.31 |
| | *Madhuca sericea* | Sapotaceae | 23.12 | 3.18 |
| Pulau Perhentian | *Vatica* sp. | Dipterocarpaceae | 49.71 | 20.36 |
| | *Buchanania arborescens* | Anacardiaceae | 31.79 | 13.02 |
| | *Syzygium cerina* | Myrtaceae | 29.34 | 12.02 |
| | *Syzygium* sp.5 | Myrtaceae | 18.15 | 7.44 |
| | *Symplocos adenophylla* | Symplocaceae | 17.48 | 7.16 |

The list of the five species that contributed to the highest amount of above-ground biomass is shown in Table 5. In Setiu Wetlands, *Melaleuca cajuputi*, *Rhizophora apiculata*, and *Alstonia spatulata* showed the highest amount of total above-ground biomass, with each contributing 84.88 t/ha (64.83%), 55.70 t/ha (80.17%) and 87.27 t/ha (58.97%) due to their abundance and dominant in each forest types formation. In Pulau Bidong, *Syzygium cinereum* contributed the most, comprising 26.49 t/ha (11.99%). *Shorea glauca* was the highest in Pulau Redang, with 532.27 t/ha (73.10%). In Pulau Perhentian, *Vatica* sp. contributed the most with an estimated total above-ground biomass of 49.71 t/ha (20.36%).

## 4. Discussion

Generally, the distribution of plants and vegetation in Malaysia is influenced by the climate, soil, and soil water [26], as well as habitat. Our study shows that the diversity of tree species in the lowland dipterocarp forests in the three major islands of Terengganu was higher than that of the edaphic forest types in Setiu Wetlands. The *Melaleuca* swamp, freshwater swamp and mangrove forests are among the various edaphic vegetation formations found in Malaysia, in which species composition is greatly influenced by adaptation to soil conditions [27]. Setiu Wetlands shows disparities in vegetation types, tree abundance, and diversity due to different BRIS soil characteristics and series that may explain the plant distribution. The BRIS soil is classified into two orders, namely entisol and spodosol; the former can be described as young and sandy soil, which is found in areas close to the sea while the latter is acidic soil combined with a mor-humus (acidic humus), which can be found more to the inland areas [28]. Several studies have been carried out on the physical characteristics of BRIS soil in the coastal area of Terengganu, which further characterized the soil order into four series: Baging, Rhu Tapai, Rudua, and Jambu [29,30]. These soil series are generally lacking in selected mineral nutrients, have low water retention capacity and are poorly structured, that limit the ability of plants to grow [22,23,29,30]. Therefore, only certain locally adaptable plants with specific soil tolerance can survive and live within these sites [3,6].

The tree density, basal area and H' value recorded in mangrove forest for this study was slightly lower but within a similar range when compared to the other regional studies, e.g., Ayer Hangat Forest Reserve, Matang Forest Reserve, Tok Bali Forest Reserve, and Kisap Forest Reserve in Peninsular Malaysia [31–34], Bahile Mangrove Forest, the Philippines [35] and in Trang Province, Southern Thailand [36] (Table S1). Our study indicated that the mangrove forest around the area was subjected to higher disturbance and experiencing low regeneration, as inferred from the low tree density and small basal area. Various studies have reported that mangrove forests are threatened by land use changes worldwide, e.g., [37–39]. In Setiu Wetlands, mangrove forests are subjected to anthropogenic disturbances such as land reclamation for development, deforestation mainly for charcoal and firewood and aquaculture activities such as shrimp farming and fish ponds [3]. These observations highlighted the urgent need for effective conservation management in the Setiu Wetlands. According to Ashton et al. [40], mangroves can regenerate productively and sustainably when given conducive conditions for stock trees to produce propagules or seeds. Meanwhile, onsite protection by law and continuous monitoring revealed higher mangrove tree regeneration survival after disturbances at another Malaysian mangrove site [34]. In terms of species diversity, the low H' value was expected due to the lack of species variation within the mangrove stands. Various comparative studies also concluded that mangrove forests had lower diversity because of their unique abiotic adaptive requirements and stand formation compared to the other tropical terrestrial forest ecosystems [32,34,41].

The freshwater swamp forest types are generally less studied in Southeast Asia when compared to mangroves [36,42,43]. A study in the freshwater swamp forest of Otuwe, Nigeria demonstrated similar findings where the H' value ranged from 0.98 to 2.13 [44]; this study recorded the H' value of 1.85. The tree density and basal area recorded for this study are slightly lower when compared to Igu [44]. This was attributed to the low tree diameter class distribution as most of the Setiu freshwater swamp trees belonged

to the small-to-medium diameter class distribution, while Igu [44] recorded more trees with large diameter sizes. The species important value ($IV_i$) shows that swamp forests are usually dominated by a few tree species [45], which agrees with the results of other studies, e.g., [32,35,36,44,46].

Our study found that most of the trees enumerated in Setiu Wetlands were in 6–25 cm DBH, and large trees greater than 25 cm DBH were rarely seen. Similarly, on the dune landscape in Jambu Bongkok, Terengganu, 433 out of 451 individuals were in the lowest diameter class of 5–15 cm DBH [18]. Poor nutrient soil properties significantly affect tree growth and diameter and may have contributed to fewer trees with large diameters greater than 25 cm [47]. On the other hand, among all the study sites, only in Pulau Redang and Pulau Perhentian, trees were seen in more than 45 cm DBH and above, e.g., *Shorea glauca* from the family Dipterocarpaceae. While in Pulau Bidong, none of the Dipterocarpaceae family was found, due partly to the impact of Vietnamese refugees that settled on the island in the 1980s and harvested most of the trees for building materials [48]. The highest numbers of trees recorded in the islands were below 24.9 cm DBH, indicating one of the characteristics of dipterocarp forest [49,50]. The distribution pattern resembled the inverted 'J' that represents the decrease in the number of individuals as the DBH of trees increases. The finding suggests that the forests are in regenerating phase due to the abundance of the understory layer [51]. Similar tree distribution is also observed in previous studies, e.g., the primary lowland dipterocarp forest in Pasoh Forest Reserve, Negeri Sembilan [52], and the lowland dipterocarp forest in Bukit Belata, Selangor [53].

Our study shows that the stem density varies according to the forest types (Table 1) and corroborates previous studies carried out in Malaysia, with a range of 800–2200 Ind/ha. For example, a total of 796 Ind/ha were recorded in the swamp forest in Sugut Forest Reserve, Sabah, 1113 Ind/ha in Tanjung Tuan Forest Reserve, Negeri Sembilan, 1576 Ind/ha in Timun Island, Pulau Langkawi, 1670 Ind/ha in the coastal forest of Terengganu, 1840 Ind/ha in Pulau Singa Besar, Pulau Langkawi, and 2200 Ind/ha in the inland forest of Pulau Redang [12,54,55]. Overall, the total basal area recorded in the three islands from 24–52 $m^2$/ha is consistent with data obtained in other studies focusing on the lowland dipterocarp forest. Generally, the total basal area is within the range of 28–52 $m^2$/ha [56], e.g., Nizam et al. [57] reported a total basal area of two forest plots established in Kenong Forest Park were 26.91 $m^2$/ha and 29.23 $m^2$/ha. Moreover, the lowland dipterocarp forest harbors a higher amount of above-ground biomass than the other forest types. Similar studies reported a range of 107.5–955.61 t/ha for total above-ground biomass [12,15,16,25,55,58–60].

Towards this point, our studies successfully assessed the forest stand structure and evaluated the level of species composition, distribution and diversity within the covered area. However, long-term monitoring of the forests may be initiated to understand more of the forest dynamics that attempt to relate present community patterns to past events and predict the outcomes of future ones concerning suitable environmental parameters such as climate and soil properties. The study's findings are much needed to enhance management practices, concurrently conserve forest resources, and guide and inform forest management activities and assessment. Furthermore, this information may contribute to the conservation, biodiversity assessment, and sustainable forest management strategies for Setiu Wetlands and islands of Terengganu.

**Supplementary Materials:** The following supporting information can be downloaded at: https://www.mdpi.com/article/10.3390/agronomy12102380/s1, Figure S1: Sample-based rarefaction curve for *Melaleuca* swamp forest, freshwater swamp forest and mangrove forest in Setiu Wetlands. Figure S2: Sample-based rarefaction curve for Pulau Redang, Pulau Perhentian, and Pulau Bidong. Table S1: Stem density, basal area, and diversity in mangrove forest for this study compared with other studies in Southeast Asia.

**Author Contributions:** Planning and designing of the research and draft of the manuscript: E.P., M.R.S. and G.E.L.; fieldwork and data curation: E.P. and M.R.S.; identification and statistical analyses: E.P. and M.R.S.; data analyses: E.P. and G.E.L.; funding acquisition: M.T.A.; writing—original draft:

E.P. and G.E.L.; writing—review and editing: M.R.S., J.M.S., K.H.L., J.W.H.Y. and M.T.A. All authors have read and agreed to the published version of the manuscript.

**Funding:** This research was funded by Trans-disciplinary Research Grant Scheme (TRGS, 2015/59373), Niche Research Grant Scheme (NRGS/2015/5313/2), Geran Galakan Penyelidikan (GGP/68007/2014/127), and Dana Pembangunan Geopark Kenyir (GEOPARK/2015/53167/3) led by M.T. Abdullah and colleagues. The APC was funded by the Research Management and Innovation Centre (RMIC), Universiti Malaysia Terengganu.

**Institutional Review Board Statement:** Not applicable.

**Informed Consent Statement:** Not applicable.

**Data Availability Statement:** Not applicable.

**Acknowledgments:** We would like to thank the Universiti Malaysia Terengganu for the permission and administrative and logistic support in conducting this study since 2015, the Forestry Department of Peninsular Malaysia, and the Setiu District Office. The second author is grateful to Jarina Mohd Jani for her insightful discussion on the local ecological knowledge about sustainable harvesting in Setiu Wetlands.

**Conflicts of Interest:** The authors declare no conflict of interest.

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
