# Peer review of "Species Composition, Diversity, and Biomass Estimation in Coastal and Marine Protected Areas of Terengganu, Peninsular Malaysia"

_agronomy, doi:10.3390/agronomy12102380_

Round 1

Reviewer 1 Report

An interesting study on the diversity and structure of the protected areas of Malaysia. I felt there were some sections that could be more succinctly summarised into tables, or even just refer to existing tables without detailing all the numbers again. Otherwise, all seems ok.

Please see the attached pdf with marked up suggestions.

Author Response

Thank you for reviewing our study. Please see attached file for our reply. 

Reviewer 2 Report

Comments to editor:   

This article presented Species composition, diversity and biomass estimation in coastal and marine protected areas of Terengganu, Peninsular Malaysia. The study is well organized and data is well arranged. The findings would be helpful for future studies. Before recommending this article for publication, there are some shortcomings for that should be resolve.   

Comments to author   

This article presented Species composition, diversity and biomass estimation in coastal and marine protected areas of Terengganu, Peninsular Malaysia. The study is well organized and data is well arranged. The findings would be helpful for future studies. Before recommending this article for publication, there are some shortcomings for that should be resolve.    General comments  Overall, the study is well designed and presented in a good way, but mostly the literature is not cited. Grammatical and typos must be revised  

Abstract 

Methods are not well presented in the abstract.   

The abstract section highlighted main results but methods are poorly described.  

Introduction The introduction part is well written but still some details are required. The authors should provide details of the significance of the protected areas by specifically focusing on the study area. What is the importance and significance of the coastal areas in shaping species composition?  How biodiversity and can protect coastal areas. Advance techniques of estimation of diversity and biomass.  

Methods and results are well organized. However, some methods are not fully described should be elaborated. Also add important references in methods and discussion section    

Conclusion is well presented. However, future recommendations based on the obtained results must be added in the conclusion section.

Author Response

(The authors gave the same response as above.)

Reviewer 3 Report

Based on Title, Abstract and Content, this manuscript is not appropriate for the Journal "Agronomy", as there is no any single component in this manuscript that is distantly related to the subject "Agronomy".

Author Response

Based on Title, Abstract and Content, this manuscript is not appropriate for the Journal "Agronomy", as there is no any single component in this manuscript that is distantly related to the subject "Agronomy".

First of all, thank you for reviewing our study. We agree with you that the journal Agronomy is not suitable to our manuscript. However, we are submitting the manuscript to a special issue of Agronomy dedicating to Recent progress to plant taxonomy and floristic studies. Below is the scope of the special issue:

Taxonomy, floristic studies, biosystematics, classical taxonomy, and modern taxonomic advances, phylogenomics, phylogenetics and biogeography, including phylogeography, and the description of well documented new taxonomic taxa, monographs, taxonomic revisions. It incorporates data from classical morphology (including both macro and micro morphology), molecular study, anatomy and ecology, distribution, molecular evolution, evolutionary development, population biology, conservation biology, evolutionary ecology, paleobiology, and related methods and theories in recent development in systematics and floristic studies. This issue invites research articles, review articles, taxonomic revisions, and short communications.